# Biogenic Preparation of ZnO Nanostructures Using Leafy Spinach Extract for High-Performance Photodegradation of Methylene Blue under the Illumination of Natural Sunlight

**DOI:** 10.3390/molecules28062773

**Published:** 2023-03-19

**Authors:** Mansab Ali Jakhrani, Muhammad Ali Bhatti, Aneela Tahira, Aqeel Ahmed Shah, Elmuez A. Dawi, Brigitte Vigolo, Ayman Nafady, Lama M. Saleem, Abd Al Karim Haj Ismail, Zafar Hussain Ibupoto

**Affiliations:** 1Institute of Physics, University of Sindh, Jamshoro 76080, Pakistan; 2Institute of Environmental Sciences, University of Sindh, Jamshoro 76080, Pakistan; 3Institute of Chemistry, Shah Abdul Latif University, Khairpur Mirs 66111, Pakistan; 4Department of Metallurgy, NED University of Engineering and Technology, Karachi 75270, Pakistan; 5Nonlinear Dynamics Research Centre (NDRC), Ajman University, Ajman P.O. Box 346, United Arab Emirates; 6Institute Jean Lamour, Université de Lorraine, CNRS, Institut Jean Lamour (IJL), F-54000 Nancy, France; 7Department of Chemistry, College of Science, King Saud University, Riyadh 11451, Saudi Arabia; 8Biomolecular Science, Earth and Life Science, Amsterdam University, Kruislaan 404, 1098 SM Amsterdam, The Netherlands; 9Institute of Chemistry, University of Sindh, Jamshoro 76080, Pakistan

**Keywords:** leafy spinach extract, ZnO nanostructures, methylene blue, scavenger

## Abstract

To cope with environmental pollution caused by toxic emissions into water streams, high-performance photocatalysts based on ZnO semiconductor materials are urgently needed. In this study, ZnO nanostructures are synthesized using leafy spinach extract using a biogenic approach. By using phytochemicals contained in spinach, ZnO nanorods are transformed into large clusters assembled with nanosheets with visible porous structures. Through X-ray diffraction, it has been demonstrated that leafy spinach extract prepared with ZnO is hexagonal in structure. Surface properties of ZnO were altered by using 10 mL, 20 mL, 30 mL, and 40 mL quantities of leafy spinach extract. The size of ZnO crystallites is typically 14 nanometers. In the presence of sunlight, ZnO nanostructures mineralized methylene blue. Studies investigated photocatalyst doses, dye concentrations, pH effects on dye solutions, and scavengers. The ZnO nanostructures prepared with 40 mL of leafy spinach extract outperformed the degradation efficiency of 99.9% for the MB since hydroxyl radicals were primarily responsible for degradation. During degradation, first-order kinetics were observed. Leafy spinach extract could be used to develop novel photocatalysts for the production of solar hydrogen and environmental hydrogen.

## 1. Introduction

In recent decades, the rapid growth of metropolitan cities has been known to be highly beneficial to human society but, at the same time, extremely detrimental to human health. Water resources are notoriously polluted due to the direct or indirect disposal of a wide range of hazardous waste materials, and this will pose a significant threat to having easy access to clean water in the coming decades [1,2,3,4]. The industrial sector produces hundreds of different products, including printing inks, pulp, paints, and plastics, all of which are produced using a large number of synthetic dyes. Some of these organic dyes are widely used in the textile industry and thrown into water streams without prior treatment [5,6,7,8,9]. In addition to organic dyes and insecticides, industrial wastewater contains toxic compounds. Color is an intrinsic property of organic dyes that has a strong impact on water bodies as it limits the photons of light and lowers the amount of dissolved oxygen. In aquatic life, these toxic aspects of organic dyes cause chronic health issues and environmental problems. Taking out dye pollutants from industrial wastewater has been gaining intense attention worldwide as a means of maintaining a green environment and enhancing the existence of living organisms. Numerous techniques have been developed to accomplish this goal, such as adsorption, biodegradation, and heterogeneous photocatalysis [10,11]. Nanostructured materials have recently been found to be effective protocols for removing organic dyes from wastewater due to their low dimensions and high surface-to-volume ratio. In addition, specific affinity-carrying agents further improve the surface effectiveness of removing dyes [12,13]. Consequently, a series of functional nanostructured materials have been reported as photocatalysts, including ZnO [14,15], Fe_2_O_3_ [16], TiO_2_ [17], and WO_3_. A number of strategies are being developed to overcome the challenges associated with wastewater treatment in order to photodegrade these dyes from aqueous solutions.

Among the various designs of photocatalysts, semiconducting photocatalysts have proven to be highly effective. Research on the proposed project aims at enhancing photocatalysis through the development of new classes of materials or the modification of surfaces. Therefore, a great deal of research has been conducted regarding the development of active photocatalytic materials [18,19,20,21,22,23,24,25,26,27,28,29,30,31]. In comparison to other metal oxide semiconductors, titanium dioxide (TiO_2_) is effective in removing dyes from water effluents, but it has been well-explored to synthesize inexpensive ZnO photocatalysts for removing synthetic organic dyes from aqueous solutions [32,33,34]. ZnO has a wide bandgap (3.37 eV) and high electron mobility (60 meV) owing to its high electron-hole binding energy. Among its many properties, ZnO is also well-known for its thermal, mechanical, piezoelectric, as well as optoelectronic properties. It has been reported that ZnO nanostructures have demonstrated outstanding photocatalytic and quantum properties over TiO_2_ nanostructures which are commonly used in photocatalysis [35,36,37,38]. While physicochemical techniques are widely used for the synthesis of ZnO, these methods are limited by the use of costly and hazardous chemicals. Meanwhile, the green preparation of ZnO nanostructures is gaining a lot of popularity among researchers as it is eco-friendly, simple, environmentally friendly, and can be scaled up, making it an effective method for removing organic dyes from industrial wastewater effluents. In green preparation methods, there is no use of organic solvents, surfactants, or other chemicals. It has been demonstrated that the green synthesis of ZnO nanostructures using plant extracts allows for the retention of the crystal phase and allows for high surface modification in order to achieve efficient photocatalytic applications [39,40]. Due to this, green preparation methods can serve as a promising alternative to physicochemical methods. A large number of phytochemicals can be found in plant extracts, which are capable of acting as reducing and/or stabilizing agents during the synthesis of nanostructured materials. Alkaloids, phenols, glycosides, passifloricins, flavonoids, cyanogenic substances, polypeptides, and alpha pyrones are some of the phytochemicals that have been useful for preparing nanostructured materials [41,42,43,44]. Spinach *(Spinacia oleracea)* is an edible green vegetable that consists primarily of leaves. It belongs to the *Amaranthaceae family*. The growth of spinach generally takes place once a year, and it is a highly valued food constituent that contains antioxidants and anti-cancer properties. Spinach primarily contains vitamins A, C, K, and folate, as well as fiber, calcium, iron, and potassium Researchers have used spinach to grow ZnO nanostructures and study their applications [45,46,47,48].

Moreover, spinach was used for its abundant crop production around the world, and we used the green leaves of spinach to collect the extract. There is no report on the transformation of one dimensional (ID) nanorod of ZnO into cluster formation using leafy extract of spinach due to the presence of various phytochemicals. These phytochemicals offer the properties of a stabilizing agent, capping agent, chelating agent, and reducing agent, which together play a vital role in tuning the surface properties of ZnO. These features of the extract of leafy spinach have never been highlighted and investigated for photocatalytic applications. As a consequence of these previous studies involving the preparation of ZnO from spinach leaf extract, it is evident that the spinach-derived ZnO nanostructures have not been investigated for the photodegradation of methylene blue (MB) under the irradiation of natural sunlight. Hence, for the first time, we are highlighting the use of it in biogenic ZnO synthesis and its contribution to removing MB green to nearly 100% under natural sunlight.

In this study, we have used the leafy extract of spinach for the surface alteration of ZnO nanostructures during the hydrothermal process. The ZnO nanostructures were investigated with respect to morphology and crystal quality. The leafy spinach extract added a vital role in the enhancement of the photocatalytic activity of ZnO nanostructures against methylene blue (MB) in an aqueous solution.

## 2. Results and Discussion

### 2.1. Crystal Arrays, Morphological Studies of Surface Modified ZnO Nanostructures

Figure 1a shows the measured XRD diffraction patterns of ZnO nanostructures prepared from spinach leaf extracts. Various ZnO samples were analyzed by powder XRD, and several crystal lattice planes were observed (100), (101), (102), (110), (103), and (112), which were well-confirmed by the standard (JCPDS card no: 79-2205). In Figure 1b, XRD suggests the presence of the hexagonal phase of ZnO that has a typical Wurtzite structure. However, phytochemicals present in spinach leaf extract have shifted their two theta angles towards a higher angle. It is evident from the crystallographic features of pristine ZnO and spinach leaf extract-assisted ZnO samples that their crystal systems are identical. With the use of significant volumes of spinach leaf extract, a significant shift has been found due to the presence of high levels of phytochemicals, so a dominant two-theta angle has been observed. The shift in the two-theta angle is higher for sample-4 compared to other samples owing to a high density of heavy biochemical molecules from the leafy extract, which might offer stress during the crystal growth. Consequently, an obvious shift in the two-theta angle was noticed for sample-4. In an intriguing development, the shift in two-theta angles may contribute to the addition of defects to the crystal arrays of ZnO, which may enhance the photocatalytic performance of these nanostructures. Based on the results of the XRD study, ZnO has been successfully synthesized using spinach leaf extract. The Scherrer equation has also been used to calculate the average crystallite size based on XRD data [49,50]. On average, crystallites are approximately 14 nm in size for sample 4, as given in Table 1. Figure 2 illustrates the measured SEM images of pristine ZnO nanostructures and spinach leaf extract-induced ZnO nanostructures for evaluating their morphology. Figure 2a illustrates the distinctive nanorod-like morphology of pristine ZnO, with a length of several microns and a diameter of 200–400 nm. In contrast, the morphology of ZnO was completely altered, resulting in thin nanosheets arranged into large clusters with a dimension of a few microns, as shown in Figure 2b–e. There is a strong indication that as spinach leaf extract volume increases, cluster size increases; however, in the large clusters, nanosheets appear to be arranged into clusters, indicating that phytochemicals play an influential role in tuning and transforming nanorod morphology into porous clusters of sheets. The shape transformation from nanorod to sheet-oriented clusters through the presence of oxygenated terminal groups of phytochemical. Therefore, these porous sheets enclosed in large clusters are very effective at removing MB dye from aqueous solutions. This is demonstrated below in the section on photocatalytic applications. Several factors determine the type of morphology transformation, including the effectiveness of the reactants, the polarized environment, hydrophobic magnetism, the synthesis mechanism, and Vander Waal forces followed by electric fields. Moreover, the energy and kinetic aspects of the preparation also affect the morphology of the nanostructured material to be grown [51,52]. After the reusability measurement for three cycles, the material stability in terms of morphology changes was studied by SEM analysis, as shown in Figure 2f. It is obvious that the material retained its shape and structure, verifying its stability for long-term reuse. Furthermore, the optical bandgap energies of pure ZnO and green synthesized ZnO samples were investigated, as shown in Figure 3. Kubelka-Maunk plots have been used to calculate the band gap of both materials [53].
(*αhʋ*)^2^ = K (*hʋ* − E_g_)

Herein, *h*—Planck constant, *ʋ*—frequency of illumination light, *α*—absorbance coefficient, E_g_—band gap, and K—proportionality constant. The calculated band gap values change with increasing the amount of leafy spinach extract. Pure ZnO has an optical band gap value of 3.10 eV, which is well in agreement with that reported in the literature. In contrast, sample-1, sample-2, sample-3, and sample-4 of the green synthesized ZnO samples have 2.90 eV, 2.82 eV, 2.71 eV, and 2.58 eV, respectively. It appears that the insertion of spinach leaf extract reduces the bandgap, which facilitates electron transportation and contributes to degradation by photocatalysis.

### 2.2. The Photodegradation of Methylene Blue under Natural Sunlight Using Leafy Spinach Extract-Assisted ZnO Nanostructures

#### 2.2.1. Effect of Catalyst Dose and Initial MB Dye Concentration

Our first step in studying the degradation of MB in an aqueous solution was to investigate the degradation of MB without the use of catalysts and under pristine ZnO illumination under natural sunlight prior to analyzing surface-modified ZnO nanostructures with spinach towards photodegradation of MB in aqueous solution. A solution of 8.22 × 10^−5^ M of methylene blue (MB) was prepared and exposed to natural daylight, and a decrease in absorbance through a UV-visible spectrophotometer was observed at various intervals of time, as illustrated in Figure 4a. MB was observed to degrade at a negligible rate without a photocatalyst. Figure 4b–d shows that both a kinetics and degradation efficiency of 5.2% could not be demonstrated by the MB under natural illumination in the absence of catalytic material. Therefore, MB requires a photocatalyst with a high degradation rate and efficiency. Based on the catalytic activity of pristine ZnO toward MB degradation, as shown in Figure 5a, we selected a catalyst dose of 15 mg. When using pristine ZnO in natural sunlight, degradation kinetics were relatively rapid. However, the degradation performance was poor because of the fast charge recombination rate of electron-hole pairs and the limited number of catalytic sites present in ZnO. Figure 5b,c illustrates the degradation kinetics governed by pseudo-first-order as the reaction rate was directly related to the concentration of MB. As can be seen in Figure 5d, pristine degradation efficiency for MB is approximately 57%. It is evident from the degradation analysis shown in Figure 4 and Figure 5 that either MB degradation in the absence of a catalyst or with pristine ZnO under the illumination of natural sunlight was insufficiently efficient. In order to achieve high-performance photodegradation of MB, a new generation of photocatalysts is urgently required. This paper introduces these challenges of mineralization of MB into harmless products. Because of its carcinogenic nature and toxic effects on the environment, we proposed the use of phytochemicals from leafy spinach extract as a method of enhancing ZnO’s catalytic performance and evaluating its role in MB dye. MB degradation was evaluated using four samples of ZnO containing different quantities of spinach extract, including 10 mL, 20 mL, 30 mL, and 40 mL. Leafy spinach extracts have been shown to alter crystal defects and the morphology of ZnO, both using XRD and SEM. As a result of using four different amounts of spinach leaf extract, we were able to confirm that ZnO-based photocatalysts with outstanding photocatalytic performance are optimized. Next, we evaluated the performance of the newly developed photocatalyst by evaluating the different catalyst doses of each sample of ZnO grown with four different volumes of leafy spinach extract, as well as the initial dye concentration and pH of the dye solution.

Figure 6a–d shows the UV-visible absorbance spectra of the 5 mg catalyst dose of four samples of ZnO, and it could be seen that the MB degradation of 8.22 × 10^−5^ M became prominent with the sample of ZnO prepared with 40 mL of leafy spinach extract. There is a linear relationship between the decrease in absorbance and the different intervals of time, as well as the absorbance decrease of the ZnO sample prepared with a higher concentration of phytochemicals from 40 mL of leafy spinach extract. In the SEM analysis, it was noted that the use of a large volume of spinach produced large clusters of ZnO with assembled nanosheets showing obvious porousness. Thus, they were more effective at removing MB dye than samples prepared with 10, 20, and 30 mL leafy spinach extracts. It is evident from the UV-visible absorbance spectra of MB in water that it has two peaks corresponding to monomers and dimers carried by MB, one at 664 nm and the other at 625 nm. As a result of the hypsochromic effect, the 664 nm peak has shown a blue shift towards lower wavelengths under natural sunlight illumination [54,55,56]. As a result of surface modification of ZnO with leafy extract of spinach, the absorbance of MB was significantly reduced after 210 min, and the intensity difference between 664 and 615 nm disappeared, suggesting that monomers degrade more rapidly than dimers. Additionally, two peaks at 664 nm were observed to be decreasing in intensity and shifting to the blue region, which was caused by the N-demethylated degradation as well as the degradation of phenothiazine. In Figure 7a–c, the relative concentrations are represented by *C_t_* and *C_0_*, where Ct is the illuminate concentration of MB with different illumination times, and *C_0_* is the dye concentration of MB in the aqueous solution at the outset. It has been shown that the photocatalytic performance of ZnO prepared with different volumes of leafy spinach extract is dominated by the irradiation time, and the rate constant is calculated using the following relation [57,58,59], which depicts the first-order reaction ln(*C_0_*/*C_t_*) = Kt.

Figure 7a,b illustrates the kinetic relationship between ln(*C_0_*/*C_t_*) and irradiation time. In the kinetic analysis, photodegradation of MB with ZnO nanostructures showed the rate constants to be approximately 3.25 × 10^−3^ min^−1^, 5.95 × 10^−3^ min^−1^, 1.31 × 10^−2^ min^−1^, and 1.42 × 10^−2^ min^−1^ for the ZnO samples prepared with 10, 20, 30, and 40 mL of spinach leaf extract. As shown in Table 2, the degradation efficiency of four samples of ZnO was also calculated, and sample-4 showed the highest degradation efficiency of 90%. It appears that the phytochemicals of leafy spinach extract contributed to the strong surface modification and tuned the optical properties of materials, which resulted in enhanced photocatalytic activity. The degradation performance against MB in 8.22 × 10^−5^ M was also evaluated by monitoring the change in absorbance spectrum after exposure to natural sunlight for varying periods of time using four samples of ZnO prepared with 10, 20, 30, and 40 mL of leafy spinach extract at 10 mg and 15 mg, respectively. Figure 8 and Figure 9 show the UV-visible absorbance spectra of MB at 8.22 × 10^−5^ M, and it could be noted that the absorbance linearly decreased with the irradiation time and the increase in the catalyst doses of various ZnO samples. For sample-4 of ZnO prepared with 40 mL of leafy spinach extract, the decrease in absorbance was more pronounced in both cases of 10 mg and 15 mg catalyst doses compared to the three other samples of ZnO prepared with 10, 20, and 30 mL volumes of leafy spinach extract. Figure 10 illustrates the reaction kinetics of MB degradation. It was found that the degradation rate increased linearly with time for both photocatalyst doses of 10 mg and 15 mg, but the degradation rate of the 15 mg catalyst dose was higher compared to the 10 mg catalyst dose at 8.22 × 10^−5^ M of MB. The removal of MB from the aqueous solution using catalyst doses of 10 mg and 15 mg yielded first-order kinetics, and the rate constants for the removal were 5.86 × 10^−3^ min^−1^, 8.72 × 10^−3^ min^−1^, 1.47 × 10^−2^ min^−1^, and 1.77 × 10^−2^ min^−1^, as shown in Table 2. The degradation efficiency was also evaluated for the photocatalyst doses of 10 mg and 15 mg for the effective removal of MB 8.22 × 10^−5^ M, and it was found to be 9.68 ×10^−3^ min^−1^, 1.12 × 10^−2^ min^−1^, 1.18 × 10^−2^ min^−1^, and 2.26 × 10^−2^ min^−1^, suggesting that the 15 mg of photocatalyst has shown a higher degradation efficiency of about 99%, as shown in Figure 10.

We have previously examined the effects of elevated concentrations of MB dye (8.1 × 10^−5^ M) on the photocatalytic effectiveness of ZnO nanostructures prepared with spinach leaf extract (40 mL). Here, we examine the effects of low concentrations of MB dye (6.21 × 10^−5^ M) on that performance. With MB dye irradiated with natural sunlight, our study evaluated the photocatalytic performance of ZnO sample-4 with catalyst doses of 5, 10, and 15 mg. In Figure 11, we show the UV-visible absorbance spectra. For each catalyst dose, the absorbance of the MB dye solution quickly decreased. However, the 15 mg catalyst dose had an excellent decrease in absorbance, indicating the superior performance activity of the catalyst in MB degradation under natural sunlight. A catalyst dose of 15 mg of sample ZnO offered large surface contact to MB molecules which involved both adsorption and oxidation simultaneously. Nevertheless, oxidation predominated due to the high density of catalytic sites provided by the newly modified ZnO nanostructures with spinach leaf extract. Similarly, the degradation kinetics were also evaluated for the MB solution using different catalyst doses of 5 mg, 10 mg, and 15 mg of sample-4, as shown in Figure 12a,b. The degradation kinetics were followed by pseudo-first-order kinetics, as shown in Figure 12a, whereas degradation is highly dependent on the MB concentration, as shown in Figure 12b. And degradation efficiency was found to be higher for the catalyst dose of 15 mg of sample-4, as shown in Figure 12c. In vitro experiments have shown that higher dye concentrations require large amounts of hydroxyl radicals in order to degrade dye molecules. However, the reactive hydroxyl species generated on the surface of the catalyst remain the same regardless of the used photon intensity, catalytic dose, or illumination time. Due to this, when the dye is used at high concentrations, the hydroxyl radicals generated are not sufficient to degrade the dye, reducing the efficiency of photodegradation.

#### 2.2.2. Effect of pH of Dye Solution on the Photocatalytic Performance of Prepared ZnO Nanostructures

The pH of the dye solution plays a determining role in the photocatalytic properties of nanostructured materials. This is because it controls the reaction kinetics of dye degradation and the production of hydroxyl radicals, which are dependent on pH [60,61,62]. In the case of sample-4, the catalyst dose was 15 mg, and the pH of the dye solution 6.21 × 10^−5^ M was adjusted between 3, 6, 9, and 12 using the range shown in Figure 13a–d. The dye solution pH affects the amount of charge flowing on the catalyst surface and the affinity of dye molecules for the catalyst surface. Based on published work, the pHpzc (point of zero charge) of pure ZnO was about 8.1, whereas the pHpzc (point of zero charge) of the ZnO-based photocatalyst prepared with leafy spinach extract was about 9.3. As it is known, the catalyst surface has a net zero charge at pHpzc. When the pH is less than pHpzc, the catalyst surface is positively charged, and when the pH is greater than pHpzc, the catalyst surface is negatively charged. The pHpzc values of pristine ZnO of 8.1 indicate that a pH lower than 8.3 would result in a surface that has a net positive charge. In contrast, a pH higher than 8.3 would result in a surface that has a net negative charge. Similarly, the ZnO sample prepared with 40 mL of spinach leaf extract has a pHpzc of 9.3, as well as a largely negative surface charge. By increasing the pH of the dye solution, the kinetics were very fast, and the removal efficiency percentage was increased, as shown in Figure 14a–c. Figure 14 illustrates how the positively protonated surface of ZnO is limited in its effectiveness in the removal of dye at low acidic pH values. On the other hand, at pH 12, hydroxyl radicals were most dominant in removing dye. This resulted in a more negative surface and an increased affinity for oxidizing cationic MB molecules, resulting in an outperforming degradation efficiency. The corresponding rate constant values for pH 3, 6, 9, and 12 were found to be 4.95 × 10^−3^ min^−1^, 1.79 × 10^−2^ min^−1^, 2.75 × 10^−2^ min^−1^, and 5.05 × 10^−2^ min^−1^ %, respectively. The overall photocatalytic performance of prepared ZnO nanostructures for fast and better visualization is summarized in Table 2. To determine the stability of the photocatalyst, we conducted a reusability test for three cycles using the green synthesized sample. As illustrated in Figure 15, the results of the reusability test are presented in barographs. It is very obvious from the bar graph (Figure 15), green-generated material performed exceptionally well against the degradation of cationic MB dye in all reusability tests. The results of this research have shown that the as-prepared samples are excellent, reusable photocatalysts that can be used in contaminated water to break down hazardous organic dyes. In addition, SEM analysis was performed to determine the stability of ZnO material in terms of shape after dye degradation, as shown in Figure 2f. Based on these results, it appears that the photocatalyst was reasonably stable and could be used for several degradation cycles.

#### 2.2.3. Scavenger Study for the Identification of Type of Radical Species Participating in Photodegradation of MB in Aqueous Solution

Using ZnO nanostructures prepared with leafy spinach extract, a scavenger study was conducted to identify the radicals predominating in the oxidation of MB, as shown in Figure 16. In addition, metal oxide nanostructures were tested as a means of inhibiting the degradation of MB 8.22 × 10^−5^ M under natural sunlight by mixing it with silver nitrate, ascorbic acid, and ethylenediamine tetra-acetate (EDTA) at a concentration of 10 mM. A dye’s degradation is either dominated by radicals like hydroxyl (·OH), superoxide (·O_2_), or by a concentration of electron-hole pairs. Specifically, it is shown that (·O_2_) and hydroxyl (·OH) radicals are involved, so we performed a scavenger study as per the published results [63,64,65,66,67], and the degradation of efficiency has been linked to the presence of these radicals [68,69]. Silver nitrate showed a large inhibition of MB degradation in this study and is known to limit hydroxyl radical production. Therefore, it was concluded that hydroxyl radical species were the main radical species accelerating the degradation of MB. The photocatalytic activity of the prepared ZnO sample with 40 mL of leafy spinach extract was compared with already existing photocatalysts, as given in Table 3. It is obvious that the activity of the presented ZnO sample is superior in terms of its low-cost fabrication, facileness, high degradation efficiency, eco-friendliness, and environmentally friendly approach to material synthesis compared to previous studies on the same topic. Hence, the presented photocatalyst can be utilized as an effective protocol for the removal of MB from industrial wastewater samples.

## 3. Materials and Methods

### 3.1. Used Chemical Reagents

The chemicals and reagents used in this study were of analytical grade without further purification. Merck (Karachi, Pakistan) provided the metal precursor zinc acetate dihydrate (ZnC_4_H_6_O_4_ of 99.9% purity) and (ammonium solution 25%) for the experiment. Methylene blue (C_16_H_18_ClN_3_S), silver nitrate, and ascorbic acid were purchased from Sigma Aldrich, Karachi, Pakistan.

### 3.2. Preparation of Leafy Extract of Spinacia oleracea

*Spinacia oleracea* was purchased from the local market in Jamshoro, Sindh. During the extraction process, *Spinacia oleracea (S. oleracea)* leaves were thoroughly cleaned with deionized (DI) water in order to remove any dirt. For the production of the paste, the preprocessed leaves (4 g wt%) were shredded into small pieces and placed in the juicer machine along with an appropriate quantity of DI water. Once the paste had been formed, it was transferred for filtration, and the filtration was repeated in order to proceed with the synthesis.

### 3.3. Green Synthesis of ZnO in the Presence of Leafy Extract of Spinacia oleracea

Our typical synthesis process included mixing already filtered leafy extract of S. oleracea (10 mL, 20 mL, 30 mL, and 40 mL) with 2.22 g of zinc acetate dihydrate into four beakers, followed by adding 5 mL of 25% ammonium solution; the specification of the sample is given in Table 4. Meanwhile, we prepared 2.22 g of zinc acetate without the leaf extract and named it pure ZnO. Moreover, each of the 5 total beakers was covered with aluminum foil and placed in a preheated electric oven at 95 °C for 5 h. Precipitations were allowed to settle at room temperature (RT) before being filtered. The developed material was washed with ethanol and DI water, dried at 120 °C for 4 h, and then burnt at 300 °C for 2 h before being converted into fine and homogeneous powder (ZnO NPs) with the help of a mortar and pestle. A brief representation of the synthesis process of ZnO nanostructures with leafy spinach extract is shown in Figure 1.

### 3.4. Characterization of ZnO Nanostructures

The pure ZnO samples and green synthesized ZnO samples were characterized for crystal structure information by recording the X-ray diffraction patterns (XRD) on the diffractometer (D8, Advance, Bruker, Germany) using the CuKα (1.542 Å) beam as the radiation source in the range of 20–80 (2θ diffraction angles). SEM morphology of all materials was examined with the Nova Nano SEM, which has a magnification range of 20–200,000× and a resolution of up to 1 nm. We measured the concentration of methylene blue dye during the photodegradation reaction using a UV-visible spectrophotometer (Lambda365, Perkin Elmer, Waltham, MA, USA) [1].

### 3.5. Photocatalytic Application of Surface Modified ZnO Nanostructures

In this study, pure ZnO and green synthesized ZnO samples were photodegraded against a cationic dye (MB) under sunlight irradiation. Before the photodegradation experiment, the initial methylene blue (MB) dye concentrations (8.22 × 10^−5^ M) and (6.12 × 10^−5^ M) were prepared, and different catalyst materials of each (5 mg, 10 mg, and 15 mg) separately were placed in the five beakers possessing 50 mL of (8.22 × 10^−5^ M) and (6.12 × 10^−5^ M) methylene blue dye at constant stirring under dark conditions for 60 min, resulting in the formation of adsorption-desorption equilibrium. It has been reported elsewhere that an average MB dye in wastewater could be found to be 2–2.28 mg/L. Furthermore, the pH of the solution mixture of MB was adjusted by adding either 0.050 M of NaOH or 0.050 M of H_2_SO_4_. Then, the solution with the photocatalyst was placed in an open atmosphere, and the solution was illuminated with sunlight, which enabled the activation of green ZnO material. Following this, 2 mL of each MB dye solution was injected into a quartz cuvette cell with a one-centimeter diameter at predetermined intervals (30 min), and the reduction in absorbance values was measured using a UV-Vis spectrophotometer. In order to determine the dye removal percentage of pure ZnO and green synthesized ZnO samples against methylene blue dye, the following universal formula was adopted:Dye removal (%) = (*C*_0_−*C_t_*/*C*_0_) × 100.

Here, *C*_0_ and *C_t_* are referred to as the initial and final concentration of MB at reaction time (*t*) after adding the catalyst, respectively.

## 4. Conclusions

In summary, phytochemicals from spinach leaf extract were used to modify the surface and morphology of ZnO nanostructures using the hydrothermal method. It was found that ZnO morphology consisted of large clusters assembled within thin nanosheets with a porous structure. XRD studies have demonstrated that ZnO has a hexagonal phase with a Wurtzite structure. Methylene blue was turned into harmless products by the use of modified ZnO nanostructures under natural sunlight illumination. According to the XRD study, the average crystallite size was 14 nm. Several parameters were considered for evaluating the photocatalytic properties of ZnO, including catalyst dose, initial dye concentration, pH of dye solution, and scavenger studies. MB photodegradation was studied using first-order kinetics. The optimal ZnO-based photocatalyst was found with a degradation efficiency of approximately 100% when using 40 mL of spinach leaf extract. In the practical applications of environmental and industrial wastewater treatment, spinach leaf extracts have shown to be a promising resource for preparing high-performance photocatalytic materials.

## Data Availability

The data presented in this study are available on request from the corresponding author.

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
