# Peer review of "Biogenic Preparation of ZnO Nanostructures Using Leafy Spinach Extract for High-Performance Photodegradation of Methylene Blue under the Illumination of Natural Sunlight"

_molecules, 2023, doi:10.3390/molecules28062773_

Round 1

Reviewer 1 Report

In this manuscript, the results of this research are conveyed thoughtfully and completely, and they are consistent with the experimental findings. However, the authors failed to explain and draw out the novelty of the work, this aspect needs to be improved. This work is worthwhile to be publish in this journal after major revision. The following issues should be addressed:

1. Introduction is well-organized but the importance and novelty of the research should be highlighted and more clearly stated. The authors should give some examples of works in the bibliography, to clear the advantage of their work in comparison with those works.

2. Maybe the author should compare their results clearly with other reported works, highlighting the advantage and disadvantages of their novel composite.

3. The authors are responsible for the English, which should be polished throughout the manuscript to clear some minor typo/grammar errors.

4. Introduction part, if possible, some important and relative reports references could help:

https://doi.org/10.1007/s10904-022-02389-8

https://doi.org/10.1021/acsomega.1c03735

https://doi.org/10.1016/j.heliyon.2022.e09652

5. More information on the light source that the author used in the study and the filter used.

6. Experimental part. Please indication initial concentration of dye. And what is the average concentration of dyes in wastewater that should be clean up?

7. Authors did not performed experiments on water purification using real wastewater. It is recommended to performed experiments on real wastewater, since there are many components that can significantly affect both catalytic properties and contaminate the catalyst.

8. Authors should discuss how prepared composites can be used in real experiments. Сan the composite contaminate water, and does it make it dangerous for human consumption. How can the composite be removed from the water after purification?

9. What is the optical band gap of the prepared material? Experimental evidences are needed?

10. What are the degraded products? An appropriate reaction mechanism of the degraded products should be discussed.

11. Stability tests are very important for any material performing as a photo-catalyst. Any morphological robustness, chemical compositional or oxidation state changes occur after photo-catalysis or not? Need experimental evidences in support of stability.

Author Response

Response to reviewer #1

We are thankful to the reviewer for useful comments and suggestions in improving the quality of manuscript prior to publication.

Comments and Suggestions for Authors

In this manuscript, the results of this research are conveyed thoughtfully and completely, and they are consistent with the experimental findings. However, the authors failed to explain and draw out the novelty of the work, this aspect needs to be improved. This work is worthwhile to be publish in this journal after major revision. The following issues should be addressed:

  1. Introduction is well-organized but the importance and novelty of the research should be highlighted and more clearly stated. The authors should give some examples of works in the bibliography, to clear the advantage of their work in comparison with those works.

Ans. We thank the reviewer for his comments. There is no report about the transformation of one dimensional (ID) nanorod of ZnO into cluster formation using leafy extract of spinach due to the presence of various phytochemicals. These phytochemicals have properties of stabilizing agent, capping agent, chelating agent, and reducing agent which together play a vital role in tuning the surface properties of ZnO. These features of leafy extract of spinach have never been highlighted and investigated for the photocatalytic applications. As a consequence of these previous studies involving the preparation of ZnO from spinach leaf extract, it is evident that the spinach derived ZnO nanostructures have not been investigated for the Photodegradation of methylene blue (MB) under the irradiation of natural sunlight. Hence, for the first time, we are highlighting the use of it in biogenic ZnO synthesis and its contribution to removing MB green to nearly 100% under natural sunlight. In the revised draft of manuscript, novelty of work is also added.  

Maybe the author should compare their results clearly with other reported works, highlighting the advantage and disadvantages of their novel composite.

Ans. We thank the reviewer for his comments. A comparative Table is provided in the revised draft of manuscript and highlights about the importance and advantages of presented work are added.

  1. The authors are responsible for the English, which should be polished throughout the manuscript to clear some minor typo/grammar errors.

Ans. We thank the reviewer for his recommendation. The revised draft of manuscript is highly polished with English and avoided typos and grammatical  mistakes

  1. Introduction part, if possible, some important and relative reports references could help:

https://doi.org/10.1007/s10904-022-02389-8

https://doi.org/10.1021/acsomega.1c03735

https://doi.org/10.1016/j.heliyon.2022.e09652

Ans. We thank the reviewer for the suggestions. The suggestions of the reviewer has been considered. In the introduction, these citations are added

  1. More information on the light source that the author used in the study and the filter used.

Ans. We have used direct natural sunlight, hence there was not any filtered light used for the degradation process;

  1. Experimental part. Please indication initial concentration of dye. And what is the average concentration of dyes in wastewater that should be clean up?

Ans. This is bit hard to say exactly the presence of MB dye in industrial wastewater because it varies from industry to industry like paper industry, textile, industry, etc and also region to region and there are together different standards of use of MB around the world. However, it has been reported elsewhere an average MB dye in waste water could be found 2-2.28 mg/L. In the revised draft these information are added

  1. Authors did not performed experiments on water purification using real wastewater. It is recommended to performed experiments on real wastewater, since there are many components that can significantly affect both catalytic properties and contaminate the catalyst.

Ans. We thank the reviewer for the point.  Yes, we did not studied the designed ZnO photocatalyst in the real wastewater samples. The reviewer is right that real wastewater contains different impurities along with the MB dye and it surely impact on the performance of material. However, we believe that first we analyze the different impurities presence in the wastewater, then our presented photocatalyst should be tested to evaluate its performance because without knowing those impurities we would be still unclear about influence of those impurities on the performance material. At the current time, we do not have facilities to analyze the wastewater and to determine the type of impurities other than MB, hence we are helpless to do so, hence apologize for the provision of real sample waste water results in the current version of manuscript. However, we appreciate the reviewer comment and we will try to manage these aspects in our future studies.

Authors should discuss how prepared composites can be used in real experiments. Сan the composite contaminate water, and does it make it dangerous for human consumption. How can the composite be removed from the water after purification?

Ans. We appreciate the reviewer comment and it is very interesting and of course opening challenge after the successful development of new generation photocatalysts for the removable of organic dyes.  The real experiments we meant to wastewater samples for the removal of MB. Yes, reviewer is right about the toxic effects of nanoparticles, but we are first dealing one problem at the moment once, we will successful in designing low cost, facile and efficient material for the 100% removal of MB from wastewater. The next issue raised by the use of nanoparticles about their toxicity, surely we need to remove the nanoparticles from the wastewater which is completely free from the MB content, then new technologies have to be built to clean the wastewater containing those nanoparticles and other impurities especially the development of nanofilters and green membranes which can easily adsorb these nanoparticles with 100% filtration and purification.

What is the optical band gap of the prepared material? Experimental evidences are needed?

Ans. In the revised draft of manuscript, we have added the experimentally determined optical band gap and discussion is also provided.

  1. What are the degraded products? An appropriate reaction mechanism of the degraded products should be discussed.

Ans. In the revised draft of manuscript, the possible degradation mechanism is described and the end products of MB are added in the revised draft of manuscript.

  1. Stability tests are very important for any material performing as a photo-catalyst. Any morphological robustness, chemical compositional or oxidation state changes occur after photo-catalysis or not? Need experimental evidences in support of stability.

Ans. In the revised draft of manuscript,  we have studied the reusability test of our material for the illustration of its stability and we have provided the SEM image after reusability test for the verification of stability of material.

Reviewer 2 Report

The manuscript provides study of the ZnO nanostructures biogenic prepared using spinach leafy extract. I have several comments as follows:

 1.      The references are obsolete considering fast developing of the materials. Please add references form the year 2022 and 2023 to create clear image of the place of the current work in the world literature.

2.      The abstract is too long. It should be shortened to clearly reflect the goal and the results obtained.

3.      In the Introduction, I propose to improve the last part to clearly underline the novelty of the current work and to emphasize why the proposed approach is better than the existing ones.

4.      The difference between Sample-1-4 is not clearly described. I recommend creating a table where the samples name and the distinguishing factors should be listed.

5.      In the XRD part (Fig. 1), there are some other peaks than ZnO - one at about 38° and 78° for sample-3. Please comment on the other phase presence.

6.      The shift of the XRD pattern of the Sample-4 as compared to the rest of samples should clearly be explained in detail.

7.      It would be nice if you could give some explanation to the changed morphology from the rods to sheets under the spinach leafy extract.

8.      I also recommend creating some hierarchy in describing each of Sample-1-4 and its influence on MB to understand the influence of the Samples growth/distinguishing parameters on photodegradation of MB (see point 4 as well).

Author Response

Response to reviewer #2

We are thankful to the reviewer for useful comments and suggestions in improving the quality of manuscript prior to publication.

Comments and Suggestions for Authors

The manuscript provides study of the ZnO nanostructures biogenic prepared using spinach leafy extract. I have several comments as follows:

  1. The references are obsolete considering fast developing of the materials. Please add references form the year 2022 and 2023 to create clear image of the place of the current work in the world literature.

Ans. We thank the reviewer for the point. In the revised draft, we have added new citation in the introduction and draft of introduction is also updated

  1. The abstract is too long. It should be shortened to clearly reflect the goal and the results obtained.

Ans. We thank the reviewer for the comments. Length of abstract is reduced and main goals and objectives of proposed study are highlighted

  1. In the Introduction, I propose to improve the last part to clearly underline the novelty of the current work and to emphasize why the proposed approach is better than the existing ones.

Ans. There is no report about the transformation of one dimensional (ID) nanorod of ZnO into cluster formation using leafy extract of spinach due to the presence of various phytochemicals. These phytochemicals have properties of stabilizing agent, capping agent, chelating agent, and reducing agent which together play a vital role in tuning the surface properties of ZnO. These features of leafy extract of spinach have never been highlighted and investigated for the photocatalytic applications. As a consequence of these previous studies involving the preparation of ZnO from spinach leaf extract, it is evident that the spinach derived ZnO nanostructures have not been investigated for the Photodegradation of methylene blue (MB) under the irradiation of natural sunlight. Hence, for the first time, we are highlighting the use of it in biogenic ZnO synthesis and its contribution to removing MB green to nearly 100% under natural sunlight. In the revised draft, we have highlighted the novelty of work and clearly differentiated from the previous studies.

  1. The difference between Sample-1-4 is not clearly described. I recommend creating a table where the samples name and the distinguishing factors should be listed.

Ans. In the revised draft, the sample 1-4 are categorized on the basis of volume of spinach extract and presented in the table as per the reviewer comment

  1. In the XRD part (Fig. 1), there are some other peaks than ZnO - one at about 38° and 78° for sample-3. Please comment on the other phase presence.

Ans. Thank you for your comment, however the addition peaks could be connected to impurities in the sample, but we could not find the any additional crystal phase system.

  1. The shift of the XRD pattern of the Sample-4 as compared to the rest of samples should clearly be explained in detail.

Ans. In the revised draft of manuscript, the shift of XRD pattern with respect to sample 4 compare to other samples is clearly explained in the revised draft of manuscript.

  1. It would be nice if you could give some explanation to the changed morphology from the rods to sheets under the spinach leafy extract.

Ans. In the revised draft of manuscript, we have stated the shape transformation from nanorod to sheet through the presence of pH of growth solution and the other oxygenated terminal groups which altered orientation of material for the formation of sheet.

  1. I also recommend creating some hierarchy in describing each of Sample-1-4 and its influence on MB to understand the influence of the Samples growth/distinguishing parameters on photodegradation of MB (see point 4 as well).

Ans. In the revised draft of manuscript, we have provided the hierarchy for each sample 1-4 and their role towards the degradation of MB for better understanding. 

Reviewer 3 Report

This manuscript reports the photocatalytic performance of ZnO midified with spinach. An improvement of the photocatalytic performance was observed. However, the data presents in the manuscript need intensive revision, at least the Figures their captions should be reorganized.

There are also some errors in the figures, such as Figure 4c, Ct/Co >>1?

Additionally, the authors should clarify why spinach was selected. Could all the green leaves serve the same role?

The manuscript is not suitable for publication at the current form.

Author Response

Response to reviewer#3:

We are thankful to the reviewer for useful comments and suggestions in improving the quality of manuscript prior to publication.

Comments and Suggestions for Authors

This manuscript reports the photocatalytic performance of ZnO modified with spinach. An improvement of the photocatalytic performance was observed. However, the data presents in the manuscript need intensive revision, at least the Figures their captions should be reorganized.

Ans. We thank the reviewer for the comments. In the revised draft, we have reshaped the figures and figure captions

There are also some errors in the figures, such as Figure 4c, Ct/Co >>1?

Ans. This has been corrected in the revised draft of manuscript

Additionally, the authors should clarify why spinach was selected. Could all the green leaves serve the same role?

Ans. We thank the reviewer for the point. The spinach was used to its abundant crop production around the world and its leaves contain plenty of favorable molecules which can serve as reducing agent, capping agent, stabilizing agents and chelating agents. Yes, we used the leaves of green spinach, but not the stem because in both cases there would be significant difference of phytochemicals, and this has been clarified in the last paragraph of introduction.

The manuscript is not suitable for publication at the current form

Ans. We have revised our manuscript, and made the possible changes and believe that reviewer feedback would be most probably positive recommendation on the revised version of manuscript.

Round 2

Reviewer 1 Report

Accepted in the present form

Reviewer 2 Report

I am satisfied with the answers. The paper now can be accepted for publication.